# The Chemistry and Health Outcomes of Electronic Waste (E-Waste) Leachate: Exposure to E-Waste Is Toxic to Atlantic Killifish (*Fundulus heteroclitus*) Embryos

**Juliet Kelechi Igbo** [1,*] **, Lucian Obinna Chukwu** [2] **, Emmanuel Olusegun Oyewo** [1] **, Jason L. Blum** [3,†] **, Ariana Schanzer** [3] **, Isaac Wirgin** [3] **, Gabriella Y. Meltzer** [4] **, Nirmal K. Roy** [3] **and Judith T. Zelikoff** [3]

[1] Department of Biological Oceanography, Nigeria Institute for Oceanography and Marine Research, 3 Wilmot Point Road, P.O. Box 12729, Lagos 106104, Nigeria
[2] Department of Marine Sciences, University of Lagos, P.O. Box 156, Lagos 100218, Nigeria
[3] Department of Environmental Medicine, School of Medicine, New York University, 341 East 25th Street, New York, NY 10012, USA
[4] Department of Environmental Health, Mailman School of Public Health, Columbia University, 722 West 168th Street, New York, NY 10032, USA
* Correspondence: igbojulie@yahoo.com; Tel.: +234-80-3386-6414
† Current address: Product Safety Labs, 2394 US Highway 130, Dayton, NJ 08810, USA.

**Abstract:** Although there is rising global concern over the environmental, ecological, and human health risks associated with the discharge of leachates from e-waste dumpsites into the aquatic ecosystems, little is known in this research area. Thus, for this study, we first defined the chemistry of the test leachate, followed by assessment of the leachate on the development of a model aquatic organism (*Fundulus heteroclitus*) used extensively as a bioassay organism in pollution studies. Chemical analyses revealed that levels of phosphate (20.03 mg/L), cadmium (Cd) (0.4 mg/L), lead (Pb) (0.2 mg/L), and chromium (Cr) (0.4 mg/L) were higher than the 2009 US EPA and the 2009 National Environmental Standards and Regulations Enforcement Agency (NESREA) permissible limits. Polycyclic aromatic hydrocarbon (PAH) burdens were dominated mainly by the high molecular weight congeners, specifically the $\sum$4rings (73 μg/L). Total polychlorinated biphenyls (PCB) levels ranged from 0.00 to 0.40 μg/L with the $\sum$deca PCBs reaching the highest concentration. For the biological studies, *F. heteroclitus* embryos (48-h post-fertilization) were divided randomly into groups and exposed to one of six e-waste leachate concentrations (10, 1, 0.1, 0.01, 0.001, 0.0001%). Significant differences ($p \leq 0.05$) between treated and control groups were observed in standard and total length, and head size. Further analysis using Duncan's post-hoc test of multiple comparison also revealed specific differences within and between specific treatment groups. We conclude that e-waste leachate arising from indiscriminate dumping into aquatic ecosystems in Nigeria contains mixtures of toxic constituents that can threaten ecosystem and public health.

**Keywords:** e-waste; leachate; *Fundulus heteroclitus*; aquatic ecosystem; dumpsite

## 1. Introduction

Electronic and electrical waste, commonly referred to as e-waste, is one of the major environmental challenges of the twenty-first century [1,2]. E-waste contains precious and semi-precious metals such as copper (Cu), manganese (Mn), iron (Fe), and platinum (Pt), as well as toxic heavy metals including lead (Pb), mercury (Hg), cadmium (Cd), and nickel (Ni). Among other chemicals, e-waste contains persistent organic pollutants (POPs), including polybrominated diphenyl ethers (PBDEs), polycyclic aromatic hydrocarbons (PAHs), and polychlorinated biphenyls (PCBs), which can all be hazardous to human health and to the environment [3].

Nigeria is one of the major countries in Africa that receive millions of tons of exported e-waste generated globally by developed countries [4]. The southwest region of Nigeria is

home to the two largest and biggest seaports in West Africa, as well as to the two largest and most heavily trafficked electronics markets in Africa [5]. The seaports serve as major trade portals for not only Nigeria, but for the rest of Western Africa. E-waste recycling and electronic dismantling in Nigeria are carried out using non-skilled manual labor without any form of work protection, appropriate equipment, or facilities [3]. These crude dismantling activities are primarily carried out by adolescents in markets where fairly-used or secondhand/slightly functioning electronic products are sold. After physically extracting the valuable materials from e-waste using equipment like hammers and chisels, the remaining products are later dumped in close proximity to water bodies. Another common practice in Nigeria, as further described by [3], is the use of open burning and uncontrolled acid bleaching to recover materials, such as copper (Cu) and plastics from cables and printed circuit boards. These activities release large amounts of toxic contaminants into the environment which could pose serious hazards for these workers and their families, local residents, and traders living near these markets, at risk of poor health outcomes. For example, a recent study [5] reported elevated blood metal levels among teenage e-waste workers in Nigeria.

Rainfall over the many e-waste mounds, where the remains of these e-waste products have been indiscriminately dumped near waterways, results in the formation of leachate. The leachate flow-through contains high concentrations of hazardous organic and inorganic chemicals, including heavy metals and POPs [6] that can contaminate aquatic ecosystems and potentially threaten human health [5,7]. The outflow of these leachate piles serves as historical spawning habitats for several important fisheries, and the associated pollution threatens the existence of these resident aquatic organisms [8]. The early life stages of certain fish species are extremely sensitive to the toxic effects of two major classes of environmental chemicals (i.e., heavy metals and dioxin-like chemicals) found in e-waste leachates [9,10].

Many heavy metals found in e-waste and their leachates can affect the developmental processes of fish during the embryonic period, resulting in a reduction of offspring quantity and quality. A study by [6] demonstrated that Cu, Cd, and Pb, in particular, promoted developmental anomalies during organogenesis. These developmental alterations included body malformations, premature hatching, deformation, and death of newly hatched larvae. According to the same investigators, these disturbances resulted in reduced numbers and poor quality of larvae as demonstrated by small body size, high frequency of malformations, and reduced viability.

The hallmark toxicity endpoints during the early life stages of fish include: craniofacial malformations; abnormal spinal curvature; yolk-sac and pericardial edema; hemorrhage; peripheral ischemia; altered hatching rate; reduced survivorship; and reduction in growth and size [11]. Studies have shown that exposure of fish to PCBs at different developmental stages has the potential to cause population decline or extinction [10,12]. Furthermore, many of the organic contaminants found in leachate (e.g., PAHs and PCBs) are known ligands for the aryl hydrocarbon receptor (AHR) pathway [12] that have been linked to additional health effects. For example, vulnerability of lake trout to dioxin-like chemicals was linked to recruitment failure, population decline, and extinction in some of the North American Great Lakes [13]. It was also demonstrated that eggs from the flounder species *Pseudopleuronectes americanus* exposed to PCBs hatched larvae that were significantly smaller in length and weight, compared to control larvae [14].

Despite the above-mentioned hazardous impacts of e-waste on ecological and human receptors, there remain limited studies [15] investigating the effects of e-waste exposure on the aquatic environment in Nigeria. This study is of great significance and of paramount importance for revealing the toxic mixture of contaminants associated with e-waste leachate, and thus the potential dangers to ecosystems and the potential human health risk posed by indiscriminate disposal of e-waste into the aquatic environment. It also provides much needed scientific information for regulatory officials and lawmakers to develop effective policies aimed at regulating import, as well as disposal of e-waste into and around water

bodies, particularly those used by people. Moreover, results of these studies provide a basis for further toxicological assessment of other e-waste contaminated aquatic environments.

The aim of this study was to identify the leachate constituents at a selected Nigerian e-waste site near residential communities, and to determine the leachate impact on life stage development of a commonly-used bioassay aquatic organism, killifish, *Fundulus heteroclitus*, which is often used to assess aquatic contamination under controlled laboratory conditions.

## 2. Materials and Methods

### 2.1. Collection of Leachate

Leachate samples were collected from ten different areas (10 m apart) in a standing body of water 15 m away from an e-waste dumpsite in Alaba, Lagos, Nigeria (N06°27′40.5″ and E003°11′32.4″) using 1000 mL glass beakers (Figure 1). The dumpsite was situated in an electronics market, which serves as a residential area primarily for traders (0.2 km away from the Alaba Rago Stream) and is located 0.5 km away from the residential town of Mosafejo and 1 km away from Ojo Beach Resort. The dumpsite consisted mostly of dismantled television sets, computer monitors, laptops, and mobile phones. There is continuous open burning activity in the dumpsite in order to reduce the volume of e-waste. Each of the ten collection samples were combined to form a composite sample, and then filtered using Whatman 10 mm IPS phase separator filter paper [2], which separated the aqueous phase from the solvent phase.

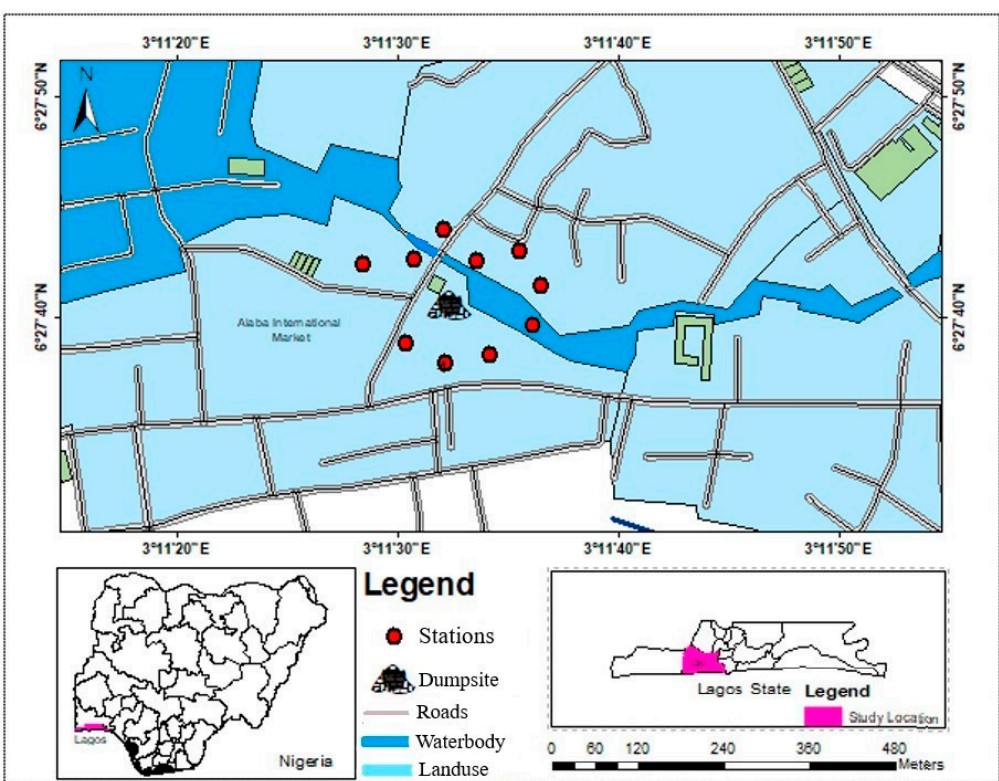

**Figure 1.** Map of the leachate collection sites surrounding a single e-waste dumpsite located 20 m away from the Alabarago stream. Leachate samples were collected from 10 different points around the dumpsite (and pooled to form a composite sample) in February (dry season) at Alaba, Lagos, Nigeria. The insets show the exact location of the dumpsite.

After filtration, the composite leachate samples were stored in 1 L glass water bottles with screw cap tops and transported for further analysis to the toxicology laboratory of the Biological Oceanography Department of the Institute for Oceanography and Marine Research, Lagos (about 106 km away from the study area) in an ice cooler for analyses. Temperature and pH were measured in situ before filtering. The filtered leachates were

then analyzed for pH, conductivity, salinity, alkalinity, and biological oxygen demand (BOD) in accordance with guidelines from the American Public Health Association [16]. All measurements were performed using a Horiba-U-10 multimeter water analyzer, while other parameters (i.e., nitrate, phosphate, and sulfate) were assessed according to [2] using a LaMotte spectrophotometer (RMN26624) employing US EPA protocols [17]. Filtered leachate was airlifted in a cooler of dry ice to the New York University Department of Environmental Medicine Laboratory, located in Sterling Forest, New York.

### 2.2. Analysis of Metals in Leachate Samples

Analysis of heavy metals in the leachate was carried out according to US EPA 2009 [17] requirements. Each 100 mL leachate sample was digested by heating with concentrated nitric acid ($HNO_3/H_2O_2$; 1:1, $v/v$), until the volume was reduced to about 3–5 mL. The digested sample was then brought up to a volume of 25 mL using ultra-pure water [18]. Concentrations of Pb, Cd, and total Cr were estimated using a Pye Unicam (Model 990) Atomic Absorption Spectrophotometer (manufactured by PG Instruments, Beijing, China).

Analysis of mercury (Hg) and arsenic (As) was carried out using a vapor method and inductive coupling plasma (ICP) atomic absorption spectrophotometer, respectively. Sample extractions were performed by measuring 100 mL of e-waste leachate in a set of digestion tubes, to which perchloric ($HC10_4$) acid, $HNO_3$, and hydrochloric acid (HCl) were added at a 1:2:2 ratio. The samples were digested at 250 °C for 1.5 h, allowed to cool to room temperature, reconstituted to a 50 mL volume using ultra-pure water, and then covered with parafilm and gently shaken. Each sample was transferred to a centrifuge tube, gently shaken for 20 min, centrifuged for 10 min (at 2800 G-force), and then transferred to a set of cold vapor vials. Analysis of the samples was then carried out using a cold vapor atomic absorption spectrophotometer interfaced with a computer system.

### 2.3. Analysis of Persistent Organic Pollutants (PAHs and PCBs) in Leachate Samples

2.3.1. Polyaromatic Hydrocarbons (PAHs)

Analysis of PAHs was carried out using methods described by [19]. Briefly, 50 mL of the composite sample was spiked with 10 μL of the PAH internal standard composed of Acenaphthene-d10, Chrysene-d12, 1, 4-Dichlorobenzene-d4, Napthalene-d8, Perylene-d12, and Phenanthrene-d10. Dichloromethane (50 mL) was then added to the sample and sonicated for 30 min. The extract was reduced to approximately 1 mL, prepared in a silica gel column (4-mm i.d. × 90 mm), and eluted with 3.5 mL of dichloromethane. The eluate was concentrated under a gentle stream of liquid nitrogen ($N_2$) to a final volume of about 100 μL. The concentration of 16 US EPA priority PAHs was measured using GC-FID Agilent 7890 A/USEPA method 522 GC/MS. The percentage recovery obtained for individual PAHs ranged from 76% to 97%.

2.3.2. Isomeric Profile of PAHs Contamination

To estimate the source(s) of PAH contamination in the leachate sample, the following four molecular indices were used:

a  Low molecular weight/high molecular weight ratio (LMW/HMW). Ratios < 1 characterize pyrolytic contamination, while ratios > 1 characterize greater petrogenic contamination [20].

b  Fluoranthene/pyrene (Fla/Py). A ratio > 1 indicates pyrolysis, while a ratio < 1 indicates petrogenic contamination [21].

c  Phenanthrene/anthracene (Ph/An) ratio. A ratio < 10 indicates pyrolysis, while a Phen/Ant ratio > 10 indicates petrogenic contamination [21].

d  Chrysene/benz(a)anthracene (BaA/Chr) ratio. A ratio value > 0.9 is associated with pyrolysis, while a ratio < 0.9 indicates petrogenic contamination [22].

2.3.3. Polychlorinated Biphenyls (PCBs)

PCBs were analyzed using the method previously described by [23]. The composite leachate sample (50 mL) was spiked with an internal standard composed of decafluorobiphenyl; ultrasonic extraction was employed to extract PCBs from the samples with 50 mL hexane/acetone (1:1 *v/v*). The extract was concentrated to approximately 3 mL using a rotary evaporator. For recovery, the sample solution was shaken with concentrated sulfuric acid ($H_2SO_4$) and, after centrifugation for 5 min (at 1792 G-force), the acid layer was discarded. This treatment was repeated several times until the hexane layer was dried with anhydrous sodium sulfate. The sample was then concentrated to approximately 1 mL for column chromatography. The concentrations of 28 PCB congeners were measured by GC-ECD Agilent 7820A. The recovery rate of individual PCB congeners ranged from 87% to 96%.

*2.4. Experimental Animals*

A total of 1000 killifish (*Fundulus heteroclitus*) embryos, at 48 h post-fertilization, were obtained from the National Oceanic and Atmospheric Administration (NOAA) laboratory in Sandy Hook, New Jersey and transported to the Department of Environmental Medicine, New York University laboratory in Sterling Forest, NY. Groups of thirty embryos were each exposed in 25 mL beakers to one of six decreasing leachate concentrations (10%, 1%, 0.1%, 0.01%, 0.001%, and 0.0001%, respectively) in five parts per thousand saltwater to mimic the seawater control [12]. A total of ninety embryos per treatment were maintained (in triplicate) in 100 mL glass beakers (at 24 °C) each using a 14:10 light-dark cycle.

The maintenance water was changed every other day using freshly prepared test solutions at the original concentrations, and individual embryos were inspected daily for viability using a light box. An embryo determined to be non-viable (opaque rather than transparent) was removed immediately from the beaker. Embryos that did not hatch at the end of the 28-day exposure period were counted as non-viable. Hatched embryos were transferred to 2 mL vials, in which they were preserved in 5% neutral-buffered formalin for later morphometric analysis.

2.4.1. Embryo Response Variables

Embryos were assessed for rate of hatchability, time for embryonic development, incubation period, and morphometric measurements of larval features. Hatchability (i.e., survival to hatching) was calculated as the proportion of viable embryos that successfully hatched from each treatment group. The embryo period duration was quantified as the time from fertilization to the day (several days later) when the first hatch was observed; the incubation period was calculated as the average total time (days) it took for each treatment group to complete hatching.

2.4.2. Morphometric Analysis

Morphometric analyses were carried out by first photographing each hatched larva using Image J software (Fiji version created by Rashand Wayne, National Institutes of Health, Bethesda, MD, USA website) to measure each individual organism. Digital images of larvae were taken at 2.5 X magnification using a Zeiss Stemi 200-C stereo microscope outfitted with a Fujifilm X-Pro1 camera. Seven measurements were obtained for each larva consisting of total length (TL), standard length (SL), head size (HS), yolk sac length (YSL), yolk sac depth (YSD), eye size-iris (ESI), and eye size-pupil (ESP) according to [12]. All measurements were in millimeters with accuracy to the nearest 0.001 mm. Figure 2 shows the precise body location where each larva was measured; all morphometric measurements were repeated three times for a given larva and the mean of the three measurements is presented.

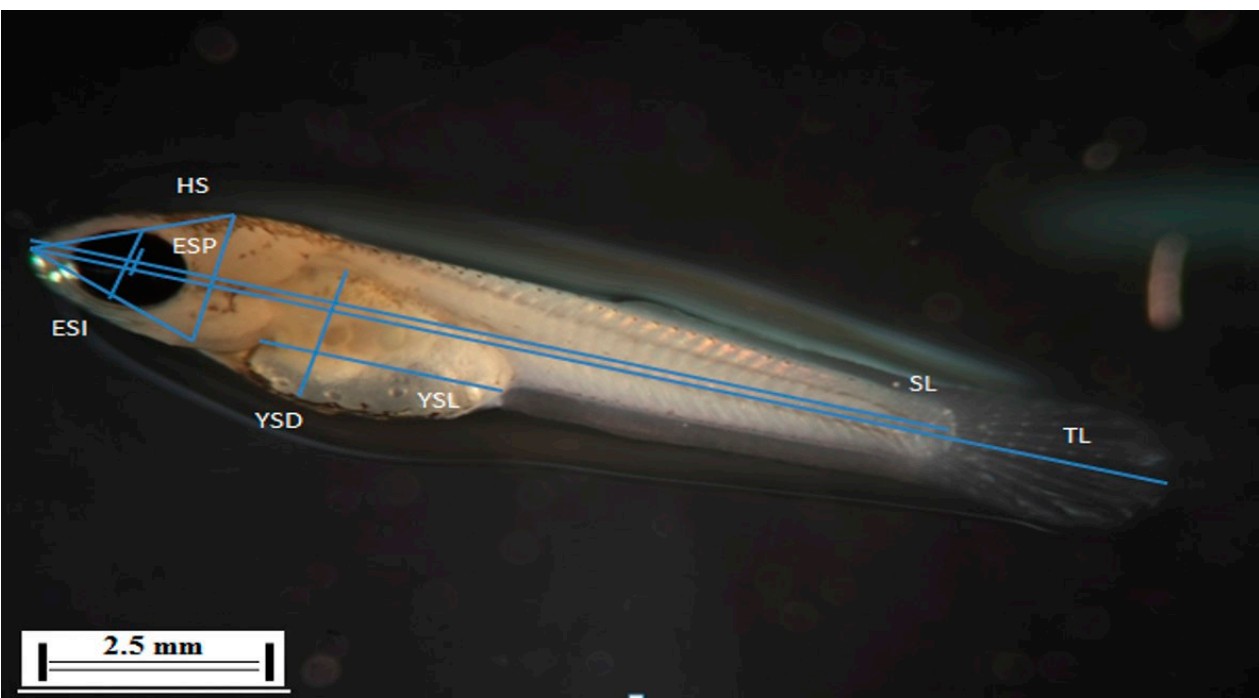

**Figure 2.** Morphometric measurements from *F. heteroclitus* larvae to assess the effects of exposure to e-waste leachate collected from Alaba, Lagos, Nigeria. The data were obtained by photographing and measuring hatched larvae. TL = total length; SL = standard length; HS = head size; YSL = yolk sac length; YSD = yolk sac depth; ESI = diameter of eye iris; and ESP = diameter of the pupil. Digital images were captured using a Zeiss Stemi 200-C stereo microscope outfitted with a Fujifilm X-Pro1 camera and lengths were determined using Image J software.

Statistics

One-way analysis of variance (ANOVA) was employed to test for significant differences ($p < 0.05$) within and between the treatment groups and control, followed by a Duncan's post hoc test of multiple comparison, when appropriate.

### 3. Results

*3.1. Physicochemical Parameters and Heavy Metal (mg·L$^{-1}$) Levels in the Leachate*

The physicochemical parameters and heavy metal concentrations of the pooled e-waste leachates are presented in Table 1. Phosphate, Cd, Pb, and total Cr levels were found to be at concentrations exceeding both the 2009 US EPA and the 2009 National Environmental Standards and Regulations Agency (NESREA; [24]) permissible levels. The pH level was within the US EPA [25] and NESREA *v/v* guidelines, while the BOD level was below permissible limits. The concentration of Cd (0.7 mg·L$^{-1}$) was in greatest exceedance of the limits, while Hg was found to have the lowest recorded concentration (0.02 mg·L$^{-1}$) of all the metals measured in the leachate.

**Table 1.** Leachate physicochemical parameters and heavy metal (mg·L$^{-1}$) concentrations.

| Parameter | Values | US EPA Standard | NESREA Standard |
|---|---|---|---|
| | | 2009 | 2009 |
| Temp | 27 °C | – | – |
| DO | 2.82 | – | – |
| BOD | 1.45 | 250 | 50 |
| Turbidity | 68.0 | – | – |
| Salinity | 0.09 | - | – |
| Alkalinity | 92 | 20 | 150 |

**Table 1.** *Cont.*

| Parameter | Values | US EPA Standard | NESREA Standard |
|---|---|---|---|
| Conductivity | 1.21 | | |
| pH | 8.05 | 6.5–8.5 | 44,721 |
| Nitrate (NO3) | 0.64 | 10 | 10 |
| Phosphate (PO4) | 20.03 * | 5 | 2 |
| Sulphate (SO4) | 17 | _ | _ |
| Lead (Pb)_ | 0.2 * | 0.02 | 0.05 |
| Cadmium (Cd) | 0.7 * | 0.01 | 0.2 |
| Chromium (Cr) | 0.4 * | 0.1 | 0.05 |
| Mercury (Hg) | 0.02 | _ | _ |
| Arsenic (As) | 0.03 | _ | _ |

Units are in $mg \cdot L^{-1}$ except conductivity ($\mu S/cm$), turbidity (NTU), salinity (ppm), temperature (°C), and pH. * Values for these parameters exceed US EPA and National Environmental Standard and Regulations Enforcement Agency (NESREA) established guidelines; BOD = biological oxygen demand.

### 3.2. Concentrations and Compositions of PAHs (µg/L) in the Leachate

Total PAH levels of individual congeners in the leachate (Table 2) ranged from 0.00 to 31.39 µg/L. The leachate was dominated mainly by the high molecular weight congeners (HMW) with $\sum$4ring congeners being at the highest concentration (83.1 µg/L). The fluoranthene (F) individual congener was measured in the leachate at the highest concentration (31.39 µg/L), followed by benzo(a)pyrene (BaP) (30.20 µg/L). Naphthalene (NAP), acenaphtylene (Acy), benzo(g,h,i)-perylene, and phenanthrene (P) were below instrument detection limits in the composite leachate sample (shown as 0 in the table). The levels of the individual congeners, chrysene (Chr), benzo(b)fluoranthene, benzo(k)fluoranthene, benzo(a)pyrene, and indeno(123)pyrene all exceeded the US EPA maximum concentration levels of 3, 2, 2, 3, and 4 µg/L, respectively.

**Table 2.** Concentrations of PAHs (µg/L) in the leachate.

| PAHs Congeners | Values | US EPA MCL |
|---|---|---|
| Naphthalene | 0 | - |
| $\sum$2rings | 0 | - |
| Acenaphthylene | 0 | - |
| Acenaphthene | 19.79 | - |
| Fluorene | 20.47 | - |
| Anthracene | 19.76 | - |
| Phenanthrene | 0 | 2 |
| $\sum$3rings | 60.02 | - |
| Fluoranthene | 31.39 | - |
| Pyrene | 18.45 | 2 |
| Benzo(a)anthracene * | 20.56 | 2 |
| Chrysene * | 12.70 | 3 |
| $\sum$4rings | 83.1 | |
| Benzo(b)fluoranthene * | 4 | 2 |
| Benzo(k)fluoranthene * | 24.96 | 4 |
| Benzo(a)pyrene * | 30.2 | 2 |
| Dibenzo(a,h)anthracene * | 15.1 | - |
| $\sum$5rings | 74.26 | |
| Indeno(1,2,3)pyrene * | 11.01 | 4 |
| Benzo(g,h,i)perylene | 0- | |
| $\sum$6rings | 11.01 | |
| Total PAHs | 228.39 | |
| LMW PAHs | 60.02 | |
| HMW PAHs | 168.37 | |
| * 7 Carcinogenic PAHs | 132.96 | |

HMW = high molecular weight ($\sum$4–6rings), LMW = low molecular weight ($\sum$2–3rings), * 7 Carcinogenic PAHs = US EPA B2 classification (as possibly carcinogenic to humans: chrysene, benzo(a) anthracene, benzo(k)fluoranthene, benzo(b)fluoranthene, benzo(a)pyrene, indeno(1,2,3) pyrene, and dibenzo(a,h)anthracene). However, the Canadian Environmental Protection Agency (CEPA) listed all PAHs as toxic substances. MCL = maximum contaminant level.

The summation of the HMW ($\sum$HMW) accounted for about 73% of the total PAH burden, while the sum of the LMW ($\sum$LMW) contributed about 26% of the total PAH concentration. The carcinogenic PAHs (cPAHs) contributed a total of 58% of the $\sum$PAH burden in the leachate. The descending order of concentration for the leachate-associated cPAHs was: benzo(a)pyrene (BaP) > benzo(k)fluoranthene (BkF) > benzo (a)anthracene (BaA) > dibenzo (a,h) anthracene (DbahA) > chrysene (Chr) > indeno (1,2,3)pyrene (i123p) > benzo (b) fluoranthene (Bbf). The PAH ring profile of the leachate (Figure 3) in decreasing order was: $\sum$4rings (36.38%) > $\sum$5rings (32.51%) > $\sum$3rings (26.27%) > $\sum$6rings (4.80%) > $\sum$2rings (not detected) in the sample.

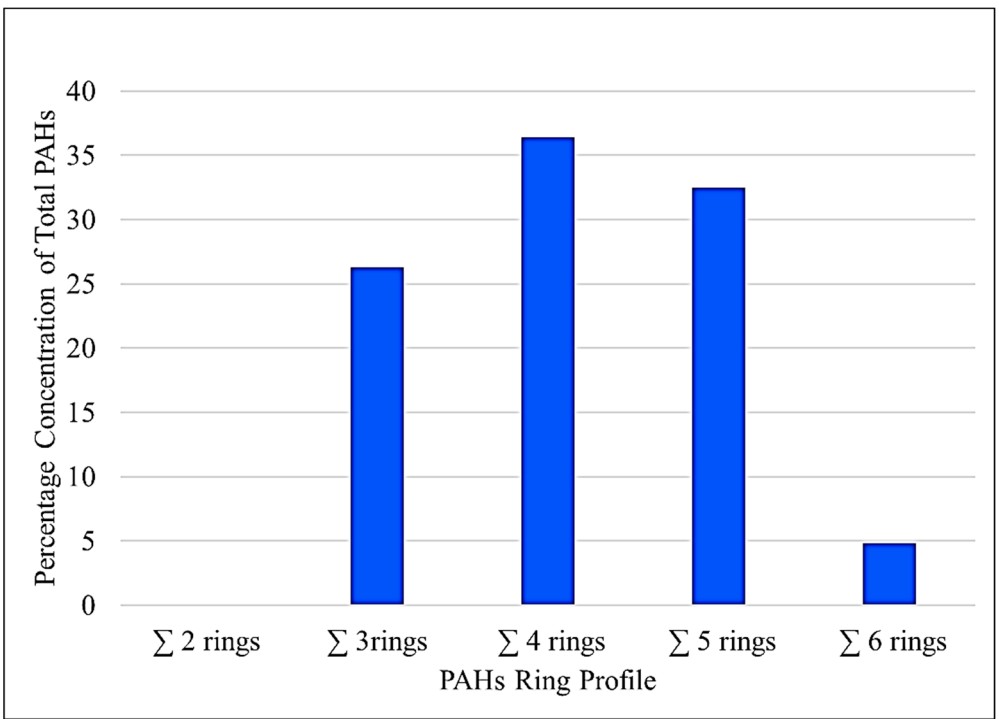

**Figure 3.** Percentage distribution of 2 to 6 ring PAH congeners to the total PAHs in the leachate sample collected from the e-waste dump site in Alaba international market, Lagos, southwest Nigeria. The 4 rings PAH congeners had the most abundant distribution in the sample leachate, while 2 ring congeners were not detected in the sample.

Isomeric Profile of PAH Contamination in the Leachate

The ratio of the molecular indices used to determine the source of PAH contamination, namely LMW: HMW compounds, was calculated to be 0.06, while that of fluoranthene/pyrene (Fl/pyr) was 1.70. The ratios of (phenanthrene/anthracene (Ph/An) and chrysene/benzo(a)anthracene (BaA/Chr)) 0 and 0.62, respectively (Table 3).

**Table 3.** Molecular indices leachate from the e-waste dumpsite in Alaba market, Nigeria.

| Isomer Ratios | Pyrolytic | Petrogenic | Leachate |
|---|---|---|---|
| LMW/HMW | <1.0 | >1.0 | 0.06 |
| Fl/Pyr | >1.0 | >10 | 1.7 |
| Ph/Ant | <10 | >10 | 0 |
| BaA/Chr | >0.9 | <0.9 | 0.62 |

HMW = high molecular weight, LMW = low molecular weight, Fl = fluoranthene, Pyr = pyrene, Ph = phenanthrene, Ant = anthracene, BaA = benzo(a)anthracene, Chr = chrysene.

### 3.3. Leachate Concentrations and Compositions of PCBs (µg/L)

The concentrations of PCBs measured in the composite leachate (µg/L) are presented in Table 4. The total leachate PCB levels ranged from non-detectable (ND) to 0.40 µg/L,

with the ∑deca-PCBs dominating (0.40 μg/L). The sum of the seven screened indicator PCBs (PCB 28, 52, 101, 118, 138, 153, 180) accounted for about 22% of the total PCB burden in the sample. Individual congeners PCB 52 and PCB 101 each had the highest concentration of 0.06 μg/L, followed by PCB 28 and PCB 153 (0.05 μg/L), while PCB 118 was not detected. The total coplanar PCBs in the sample ranged from non-detectable to 0.07 with only three congeners, i.e., PCB 105 (0.01 μg/L), PCB 114 (0.07 μg/L), and PCB 167 (0.01 μg/L), presenting levels above instrument detection limits. These particular congeners accounted for about 7% of the total PCB concentration of the test leachate.

**Table 4.** Concentrations of PCBs (μg/L) in the leachate.

| PCBs Congeners | Values |
| --- | --- |
| CB8 | 0.21 |
| ∑di-PCB | 0.21 |
| CB18 | 0.07 |
| CB28 | 0.05 |
| ∑tri-PCB | 0.12 |
| CB44 | 0.04 |
| CB52 | 0.06 |
| CB60 | 0.04 |
| CB77 | ND |
| CB101 | 0.06 |
| CB81 | ND |
| CB105 | 0.01 |
| ∑tetra-PCB | 0.15 |
| CB114 | 0.07 |
| CB118 | ND |
| CB123 | ND |
| CB126 | ND |
| ∑penta-PCB | 0.07 |
| CB128 | 0.03 |
| CB138 | 0.03 |
| CB153 | 0.05 |
| CB156 | ND |
| CB157 | ND |
| CB167 | 0.01 |
| CB169 | ND |
| ∑hexa-PCB | 0.12 |
| CB170 | 0.02 |
| CB180 | 0.02 |
| CB185 | 0.04 |
| CB189 | ND |
| ∑hepta-PCB | 0.08 |
| CB195 | 0.03 |
| ∑octa-PCB | 0.03 |
| CB206 | 0.03 |
| ∑nona-PCB | 0.03 |
| CB209 | 0.40 |
| ∑deca-PCB | 0.40 |

**Table 4.** *Cont.*

| PCBs Congeners | Values |
|---|---|
| ∑total PCBs | 1.21 |
| ∑ICES indicator PCBs | 0.27 |
| * ∑coplanar PCBs | 0.09 |

ND = not detected, Total International Council for the Exploration of the Sea (ICES) indicator PCBs = sum of concentrations of PCB 28, 52, 101, 118, 138, 153, 180. * Total dioxin-like (coplanar) PCBs = sum of concentration of PCB 77, 81, 105, 114, 123, 118, 126, 156, 157, 167, 169, and 189. USEPA MCL = maximum contaminant level for PCBs = 5 µg/L.

The composite sample consisted of ∑di to deca chlorinated homologues with PCB 209 being measured at the highest concentration of 0.40 µg/L and accounting for about 33% of the total PCB sample burdens (Figure 4). ∑-octa and ∑-nona PCBs were found at the lowest concentrations of 0.03 µg/L, with each accounting for about 2.47% of the total leachate-associated PCBs.

### 3.4. Biological Effects of E-Waste Leachate Exposure on the Embryo and Larvae of *Fundulus heteroclitus*

Concentration-dependent reduction in embryo survival was observed amongst e-waste treatment groups 1 (10%) and 2 (1%), both of which had the shortest median hatching time of 20 days (Figure 5a). Group 3 (0.1%) hatched in 23 days, while groups 4 (0.01%), 5 (0.001%), and 6 (0.0001%) had median hatching times identical to that of the control group (i.e., 22 days). However, each recorded hatching period across treatment groups was not significantly different ($p \geq 0.05$) from the control (Figure 5b).

The morphometric larval parameters of *F. heteroclitus* exposed as embryos to e-waste leachate, including total length (TL), standard length (SL), head size (HS), yolk size length (YSL), yolk size depth (YSD), eye size iris (ESI), and eye size pupil (ESP), were statistically-analyzed using a one-way analysis of variance (ANOVA). Findings revealed a significant difference ($p < 0.05$) in total length ($p \leq 0.01$), SL ($p \leq 0.01$), and HS ($p \leq 0.01$) between the e-waste treatment groups 1–3 (dilutions 10, 1, and 0.1 % respectively) and seawater controls. A further analysis using Duncan's post-hoc test of multiple comparisons also demonstrated specific differences within and between treatment groups. As shown in Table 5, TL, SL, and HS were statistically different from the control in all e-waste treatment groups, while no significant differences between or within treatment groups were observed for YSL, YSD, ESI, or ESP. Specifically, significant differences in TL were observed between e-waste leachate treatment groups 1 (10%) and 2 (1%) and groups 3 (0.1%), 4 (0.01%), 5 (0.001%), and 6 (0.0001%). Similarly, SL of treatment groups 1 (10%), 5 (0.001%), and 6 (0.0001%) were statistically different from treatment groups 2 (1%), 3 (0.1%), and 4 (0.01%). Further, HS was significantly different between treatment groups 1 (10%), 3 (0.1%), 4 (0.01%), and 5 (0.001%).

**Table 5.** Effects of e-waste on the seven morphometric larval parameters (i.e., total length (TL), standard length (SL), head size (HS), yolk size length (YSL), yolk size depth (YSD), eye size iris (ESI) and eye size pupil (ESP)) measured in *Fundulus heteroclitus*.

| TG | TL | SL | HS | YSL | YSD | ESI | ESP |
|---|---|---|---|---|---|---|---|
| TG1 | 3.47 ± 2.0 [a] | 3.43 ± 0.07 [ab] | 2.18 ± 0.09 [ab] | 1.27 ± 0.23 [a] | 1.42 ± 0.56 [a] | 1.48 ± 0.57 [a] | 2.18 ± 0.58 [a] |
| TG2 | 5.43 ± 0.38 [a] | 3.12 ± 0.12 [a] | 2.60 ± 0.08 [c] | 0.98 ± 0.06 [a] | 0.56 ± 0.18 [a] | 0.34 ± 0.05 [a] | 0.18 ± 0.09 [a] |
| TG3 | 6.40 ± 0.42 [b] | 4.20 ± 0.03 [b] | 2.07 ± 0.07 [a] | 1.16 ± 0.05 [a] | 1.07 ± 0.56 [a] | 0.72 ± 0.68 [a] | 2.18 ± 0.59 [a] |
| TG4 | 5.21 ± 0.45 [b] | 3.46 ± 0.46 [b] | 3.60 ± 0.02 [d] | 1.21 ± 0.11 [a] | 1.08 ± 0.60 [a] | 0.71 ± 0.62 [a] | 2.18 ± 0.52 [a] |
| TG5 | 6.41 ± 0.43 [b] | 3.40 ± 0.25 [ab] | 2.48 ± 0.21 [bc] | 1.21 ± 0.12 [a] | 1.48 ± 0.69 [a] | 1.13 ± 0.66 [a] | 2.18 ± 0.59 [a] |
| TG6 | 6.20 ± 0.23 [b] | 3.74 ± 0.1 [ab] | 2.54 ± 0.25 [c] | 2.54 ± 0.25 [a] | 0.75 ± 0.02 [a] | 0.36 ± 0.09 [a] | 2.18 ± 0.02 [a] |
| Control | 7.78 ± 0.05 [c] | 6.71 ± 0.05 [d] | 4.35 ± 0.12 [e] | 1.21 ± 0.33 [b] | 0.81 ± 0.07 [b] | 0.46 ± 0.08 [b] | 0.09 ± 0.02 [b] |

Values are mean ± standard deviation of triplicate determinations. Groups in the same column with similar superscripts were not significantly different at $p \geq 0.05$. TG—treatment groups.

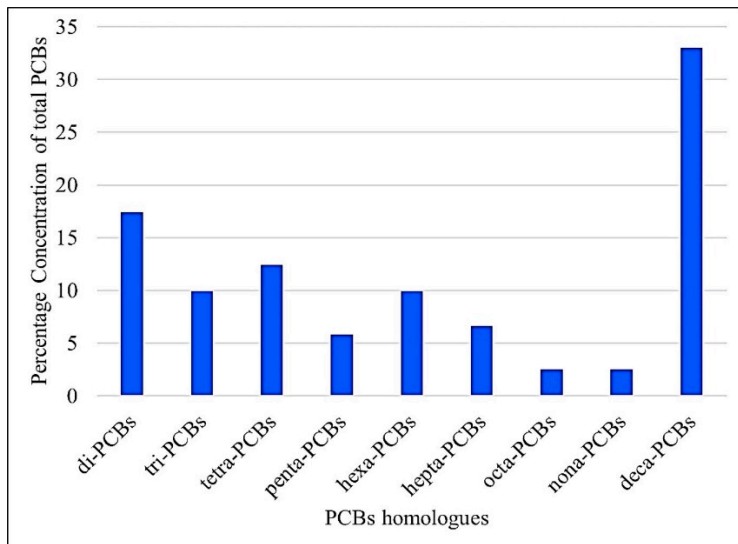

**Figure 4.** Percentage distribution of the sum of di-PCBs to the sum of deca-PCBs in the leachate sample collected from the e-waste dump site in Alaba international market, Lagos, southwest Nigeria. The deca-PCBs had the highest distribution, while the octa- and nona-PCBs were the lowest in occurrence.

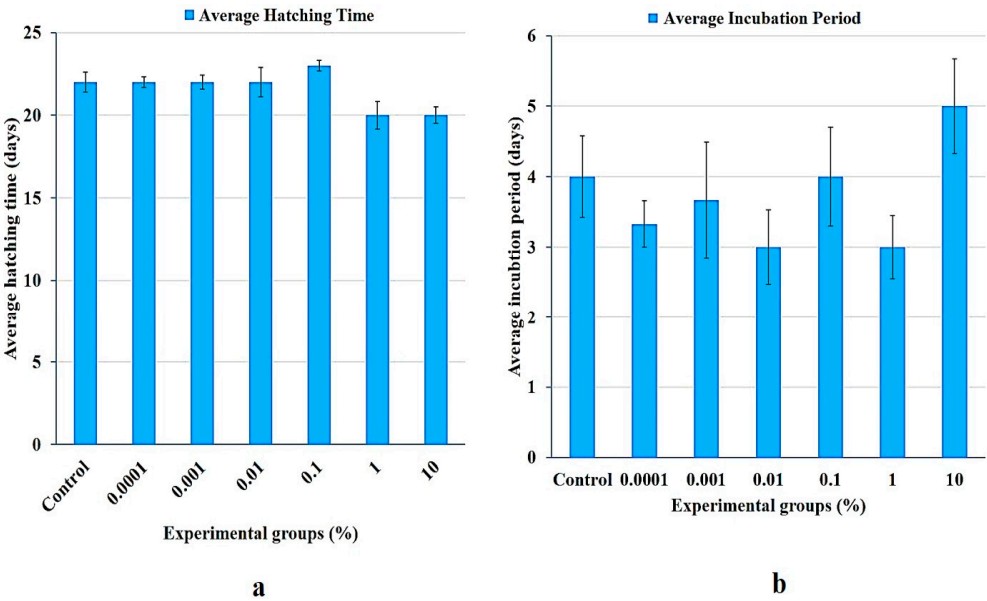

| a | b |
|:---:|:---:|

**Figure 5.** (**a**,**b**): A dose response graph of larval length. Treatment groups 1 and 2 containing the 10% and 1% e-waste leachate, respectively, show a modest (albeit, not significant) reduction in larval length, compared to the control.

## 4. Discussion

### 4.1. Leachate Contaminant Outcomes: Metals

Leaching of heavy metals from e-waste has been reported in a variety of studies [3,5,25]. In our studies, heavy metals (i.e., Hg, Cd, Cr, and Pb) measured in the leachate were above the permissible levels for leachate set by the US EPA in 2009, but below the levels previously reported by [2] in leachate collected from dumpsites in the computer village and Alaba international market, Nigeria. Such discrepancies could be due in part to the composition of materials in different e-waste dumpsites, and, thus, differences in heavy metal leachate compositions; moreover, our study utilized a composite leachate collected from different site locations compared to studies by [26]. Heavy metal contamination in leachate suggests that environmental contamination from the e-waste site migrated from the dumpsite into surrounding waters. The accumulation of end-of-life electronics and illegal recycling has

contributed to the recorded high levels of heavy metals in leachates [2]. Lead contamination in e-waste leachate, likely coming from printed circuit boards, computer monitors, and Pb acid batteries in the pile [27], is increasing over time. Exposure to increased levels of Pb emissions up to 10 kg were reported to have decreased test scores among elementary school pupils in Florida, USA [28]. Cadmium, also found at high levels in many leachate samples, is primarily released from chip resistors, semiconductors, and Ni-Cd batteries in the e-waste dumpsite [5], while total Cr levels likely leach from galvanized steel plates and hardener for steel housing used in electronic equipment [29]. High levels of Hg in leachate likely come from e-waste-associated relays, switches, and printed circuit boards [3]. Overall, the leaching of materials embedded in end-of-life electronic equipment found at the tested dumpsite or through the burning of e-waste could have contributed to the high levels of heavy metals measured in our recovered leachate sample. These findings are supported by a previous study [30], where high levels of Pb were found to have leached from computer printed wire boards and cathode ray tubes in Florida (USA) into municipal solid waste landfill leachates.

### 4.2. PAHs

The dominant constituents in the leachate tested in this study were the HMW PAH congeners, accounting for about 73% of the total PAH contamination [2]. These findings could be attributed to incomplete combustion resulting from continuous open burning of e-waste at the dumpsites [2]. The molecular indices used to determine the source of PAH contamination in the e-waste leachate (i.e., LMW: HMW, Fl/PY, P/A, and BaA/Chr) further confirm that PAH contamination came primarily from a pyrolytic source with petrogenic input. This finding could be due in part to the fact that the Lagos dumpsite selected was located close to electronics shops and residential quarters where generators are used for producing electricity [2]. The values of $\sum$PAHs analyzed in the pooled leachate sample in our study were at higher levels than those found in previously published e-waste leachate studies [2,31,32]. Amounts of carcinogenic PAHs (e.g., Chr, BkF, BbF, BaP, and I123P) analyzed in the leachate sample exceeded the US EPA maximum contaminant limit (MCL); in the absence of any Nigerian PAH standards, the US EPA MCL was used for comparisons.

### 4.3. PCBs

The $\sum$PCB concentration in the leachate sample used in this study exceeded the US EPA permissible limit of 0.5 μg/L in leachate. Contamination of the e-waste leachate with PCBs could be speculated to have arisen from di-electric fluids such as transformers and capacitators found in our tested dumpsite from used electronic equipment [3]. The $\sum$PCBs concentration observed in this study is within the range of values reported by [33] in municipal and industrial sludge from the Greater Thessaloniki area, northern Greece. However, the $\sum$PCBs measured in the pooled leachate from our study was lower than those reported in a study evaluating landfill e-waste leachate from Nairobi [33]. PCBs measured in the leachate sample used in our study consisted largely of homologues with higher chlorination levels (i.e., penta to deca-PCBs) that are capable of greater persistence in the environment. PCBs with a high degree of chlorination are more persistent in the environment than those with lower degrees of chlorination (di to tri-PCBs) [34,35]. Co-planar PCBs are also called dioxin-like PCBs (dl-PCBs) because of their co-planar structure similar to that of dioxin, and their toxicity is related to 2,3,7,8-tetrachlorodibenzo-p-dioxin [36].

### 4.4. Biological Outcomes

The reduction in embryo survivability observed in this study could be a result of any or all of the toxic constituents identified and measured in the leachate. Support for this notion comes from studies demonstrating reduced embryonic survivability in fish following exposure to e-waste constituent contaminants, including Cd and total Cr [37,38], complex mixtures of PAHs from petrogenic sources [39], and to PCBs under clean laboratory conditions [14]. Studies have also demonstrated that exposure of fish embryos to heavy

metals and POPs have adverse impacts on survival, growth rate, and quality of embryonic development, leading to larval abnormality, malformations, and increased mortality [40]. Moreover, heavy metals and POPs have been shown to induce morphological alterations and growth defects even at relatively low concentrations [37,41]. In a 2008 study [42], exposure during embryonic development of the common carp (*Cyprinus carpio*) to Cd, Pb, and Cu induced premature hatching, reduced hatchability, and impairment of embryonic development. An earlier study by [43] also demonstrated that *Cyprinus carpio* embryos exposed to Cd produced small yolk size at hatching, low survival, and reduced body length. Wirgin [11] observed that exposure to PCB 126 reduced the rate of hatching and survival of short-nose sturgeon (*Acipenser brevirostrum*), and a study by [44] linked impaired growth effects in the early stages of rainbow trout (*Onchorhynchus mykiss*) development to exposure to a mixture of PCBs (Aroclor 1260). Moreover, winter flounder (*Pseudopleuronectes americanus)* larvae hatched from eggs exposed to water from a PCB-contaminated bay were significantly smaller in length (2.96 mm) and weight (0.018 mg) compared to those collected from a clean water control site (3.22 mm, 0.022 mg, respectively) [14]. Smaller larvae are inefficient predators and more vulnerable prey due to reduced visual and swimming ability [37,45].

In contrast to the aforementioned studies demonstrating the sensitivity of larvae and/or embryos to e-waste associated contaminants, [46] reported a lack of detrimental effects on fertilization success after exposure of sheepshead minnows (*Cyprinodon variegatus*) to the PCB mixture, Aroclor 1254. A number of factors could help explain the dissimilarities observed between the studies, such as differences in fish strain, PCB exposure concentration, and/or presence of different PCB congeners.

In our study, early life stage exposure of *F. heteroclitus* to increasing concentrations of pooled e-waste leachate significantly altered TL, HS, and SL of exposed larvae. Individual or contaminant mixtures identified in the leachate recovered from a commonly exploited e-waste site in Lagos, Nigeria were likely partly responsible for the changes in embryo and/or larval parameters observed in this study. Taken together, results from our current study demonstrate that exposure of *F. heteroclitus* to e-waste leachate during an early life stage impacts embryonic development that, in turn, could lead to reduced numbers of adult fish in surrounding waters. The results of this study detail the aquatic toxicity, particularly during early life-stages of *F. heteroclitus,* of e-waste exposure, and documents the complex and highly toxic environmental contamination found in e-waste leachates associated with waterways in the Alaba international market in Lagos, Nigeria. These findings should serve as a 'wake-up' call for countries that import large amounts of e-waste (i.e., low- to middle-income countries (LMICs) such as Nigeria that relies on subsistence fishing as their primary food source.

Though fish tissue burdens were not measured in this study, it could be speculated that fish collected from nearby the tested leachate site could pose a contaminant exposure pathway for humans, leading potentially to a variety of health complications associated with individual pollutant constituents, such as impaired lung function, thyroid function, hormone expression, cognitive impairments, spontaneous abortions, premature births, reduced birthweights and birth lengths, and DNA/ genotoxic damage. Indeed, such health outcomes have been observed in e-waste contaminated communities in Nigeria and China where studies have demonstrated elevated e-waste-associated contaminants in blood, urine, hair, serum, and placenta of exposed individuals [5,9]. Future biomonitoring and epidemiological studies are critically needed to assess such health parameters in residents and workers at the e-waste dumpsite examined in this study. In a report by Sweta [47], regular aquatic risk assessment was recommended for water bodies in close proximity to e-waste processing plants to evaluate and identify potential risks posed by exposure to e-waste contaminants. Moreover, goals should be set to achieve a comprehensive and successful e-waste management program that encompasses all stakeholders, including unskilled e-waste recyclers, recycling companies, traders, general public, and regulatory authorities, for which risk communication skills are a must [48]. In general, it would be prudent for LMIC

governments who allow the import of e-waste into their countries to pass appropriate legislation that addresses the dumping of such products into nearby waterways.

## 5. Conclusions

The significant impact of e-waste leachate on exposed *F. heteroclitus* embryos is a direct reflection of the potential toxic effects that result from crude recycling activities and illegal dumping of e-waste in close proximity to waterways. The major key finding from this study is that e-waste leachate collected from the Alaba International Market consists of a toxic "soup" of carcinogenic and environmentally-persistent chemical constituents that can adversely affect ecosystem health and, potentially, pose a risk for public health. The findings in this study should serve as a wakeup call for LMIC government agencies to take immediate and preventive actions to mitigate further damage to their ecosystems. We recognize certain limitations in our study, including a small sample size and the fact that samples were pooled, rather than analyzed individually to determine potential 'hot-spots'; use of a fish model not indigenous to Nigeria; speciation of total Cr to define the amount of carcinogenic hexavalent Cr; and failure to measure metal/POP embryo burdens to define actual dose to the organism. However, despite several minor study weaknesses, this study not only provides a basis for future ecosystem and human health studies for potentially exposed people living or working around such sites, but also alerts governmental officials to the need for better prevention and intervention strategies. Future research recommendations include the frequent monitoring of all water bodies surrounding e-waste sites to ensure sustainability of the waterways and prevent potential human exposure. While we recognize the importance of e-waste for LMIC economic stability, it is critical that future studies develop strategies that can balance environmental health with financial viability.

**Author Contributions:** Conceptualization, J.K.I. and I.W.; funding acquisition, J.T.Z.; investigation, J.K.I.; methodology, I.W. and N.K.R.; resources, J.T.Z.; supervision, L.O.C., E.O.O. and J.T.Z.; writing—original draft, J.K.I.; writing—review and editing, J.L.B., A.S., G.Y.M. and J.T.Z. All authors have read and agreed to the published version of the manuscript.

**Funding:** This research was funded by NYU NIEHS Center of Excellence P30 ES000260.

**Institutional Review Board Statement:** Not applicable.

**Informed Consent Statement:** Not applicable.

**Data Availability Statement:** The data presented in this study are available in this article.

**Conflicts of Interest:** The authors declare no potential conflict of interest with respect to the research, authorship, and/or publication of this article.

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
