# Peer review of "The Chemistry and Health Outcomes of Electronic Waste (E-Waste) Leachate: Exposure to E-Waste Is Toxic to Atlantic Killifish (Fundulus heteroclitus) Embryos"

_sustainability, doi:10.3390/su141811304_

Round 1

Reviewer 1 Report

The manuscript under review is devoted to an urgent environmental problem - the impact of pollution on aquatic organisms. It should be noted that the authors consider such atypical pollutant as disposable electronic devices. The volume of such wastes has recently been growing rapidly, and there are practically noworks on studying their environmental effects. It determinesthe high relevance of the investigation. The manuscript is well structured and presented. The original results are presented. There are only a few comments.

It is necessary to improve the quality of Figure 1, make it clearer

I think, it is better to write some conclusion at the end: 2-3 sentences about the main advantages of their work and how this content will be used in future. 

Author Response

 RESPONSE TO REVIEWER 1 COMMENTS

The authors would like to thank the reviewers for their helpful comments that have improved the quality of the manuscript. Words/sentences that have been changed in the article text either for clarity or to address the reviewer’s comments, are lettered in red.

Responses to Reviewer’s Comments:

Response to Reviewer 1:

  1. The quality of Figure 1 has been enhanced for greater clarity.
  2. We have added several sentences in the discussion and in a concluding paragraph that highlights the main advantages of the study, and how the data could be used to guide policy to protect the public health.

Reviewer 2 Report

"A study by[14]..." can be better described (lines 64, 81, 94-95, etc)

zebrafish (line 97)

There are no  "physiological determinations" in this article (lines 134-135)

correct de perchloric acid formula (line 147)

"Wirgin et al. (2012)." (line 220)

F. heteroclitus non italic (line 227)

(mg/L) should be mg.L-1 (results and discussion)

The figures 1, 2, 3 are deformed

Table 5 must be corrected (superscript contol group of YSL, YSD, ESI and ESP)

Which previous report (lines 368-369)?

If is insignificant (albeit, not significantly), must not be discussed (lines 421-424)

What was relevant in this study (line 444), what about your result of yolk reserve, what does mean?

77 μg-1 (wet weight) in the eggs,"?? (line 450)

There are no supporting statistics that leachate impacted hatching time (line 455)

Format the references

Author Response

RESPONSE TO REVIEWER 2 COMMENTS

The authors would like to thank the reviewers for their helpful comments that have improved the quality of the manuscript. Words/sentences that have been changed in the article text either for clarity or to address the reviewer’s comments, are lettered in red.

Responses to Reviewer’s Comments

  • All figures in the Results Section were redone and bolded for increased clarity, as the reviewer suggested.
  • A conclusion paragraph (5 sentences) was added to the Discussion section, as suggested.
  1. study by [14] was revised on Line 71 [ Introduction section].  
  2. Lines 54, 81, 94-85, 97 were revised [Introduction section].
  3. The phrase “physiological determination” was deleted as suggested by the reviewer [Line 112].
  4. We corrected the chemical formula of perchloric acid [Line 122].
  5. The reference for Wirgin et al [2012] was corrected as suggested [12] [Line 190].
  6. 6. heteroclitus is now in italics [Line 194].
  7. The term mg/L was changed to mg.L-1 in the Result and Discussion section
  8. Table 5 Superscript in the control group of YSL. YSD, ESI and ESP was corrected.
  9. The previous report was cited and given a reference number [2] [Line 275].
  10. As suggested, we deleted all discussion concerning outcomes that did not reach statistical

               significance.      

  1.   The discussion on yolk reserve was deleted as the reviewer suggested, due to a lack of    

                relevancy for our study.

  1. Deleted the sentence referring to 77 µg-1 wet weight [Discussion section].
  2. As suggested, we deleted the discussion on hatching time, as it was not a statistically    

              significant change in response to the leachate. 14.

  1. References have been re-formatted based on the journal’s suggested format.

Reviewer 3 Report

Dear Editor, thank you very much for your invitation to review this manuscript ID: sustainability-1826148, submitted to Sustainability.

The original paper "The Chemistry and Health Outcomes of Electronic Waste (E-waste) Leachate: Exposure to 1 E-waste is Toxic to Atlantic Killifish (Fundulus heteroclitus) Embryos" by Igbo et al. provides valuable information regarding the leachate constituents in Nigeria and the impact of e-waste leachate to F. heteroclitus embryos.

From this point of view, this study intends to fill out the knowledge gaps of these current global concerns in aquatic ecosystems (marine pollution). However, I do not recommend publishing the paper at the current stage because the authors need to make a few improvements:

Quality of Structure:
There are many mistakes in the text (misspellings). I also suggest you enhance your figures because the quality is not as good as the minimum required by this journal.

Abstract
:
- I propose you rewrite the abstract section or reorganize the ideas. An excellent abstract needs to describe: The context, gap, purpose, methodology, results, and conclusions. This way, I do not find the items in a logical sequence, and you can add more connections between the ideas and outcomes. Please, could you fix it?
I strongly recommend you make changes to this abstract section.

- “Results demonstrated that 28 levels of phosphate (20.03 mg/L), cadmium (Cd) (0.4 mg/L), lead (Pb) (0.2 mg/L) and chromium 29 (Cr) (0.4 mg/L) were either higher than the 2009 United States Environmental Protection Agency 30 (US EPA) or the 2009 National Environmental Standards and Regulations Enforcement Agency 31 (NESREA) permissible limits.”. Where? It is not clearly for me.

- You are using acronyms without a previous description. “PAH” and “PCB”.

- Fix the “Key Words” to Keywords.

Introduction:

- The introduction section and aim of the paper are well written and help to clarify the main concepts in this subject.

- L53-69: Please, revise this paragraph because it is too long and contains some repeated words.

Materials and Methods:
- Definitely, the quality of figures is not good. Please, considering change images in this manuscript. I cannot read the words in figure 1.

-
The methodology is appropriate for the purpose goal and is based on official guidelines.

- L196-197: “Embryos were maintained (in triplicate) in 100 mL glass beakers (at 24°C) using a 196 14:10 light-dark cycle.”. How many embryos per treatment? You should include this information.

- The graph presented the results as mean ± standard error, right? Please, check all graphs out.

Results:
-
“Figure 5 (a and b)”. Could you add the letter “a” and “b” and standardize the drawn figures?

- Table 3 is not a table like 5, 4, 2 and 1. Considering fix this table.

- The outcomes are well shown.

Discussion:
- The discussion is well supported. However, you can include a little bit more recent references.

- Please check the scientific names throughout the manuscript.

Author Response

RESPONSE TO REVIEWER 3 COMMENTS`             

The authors would like to thank the reviewers for their helpful comments that have improved the quality of the manuscript. Words/sentences that have been changed in the article text either for clarity or to address the reviewer’s comments, are lettered in red.

Responses to Reviewer’s Comments:

Response to Reviewer 3:

  1. Quality of Structure:
  2. We have carefully reviewed the manuscript for misspellings, and now feel that all have been corrected.
  3. As we have responded to the comments of reviewer’s 1 and 2 regarding the lack of clarity in the figures, all have been redone.
  4. Abstract:
  5. The abstract has been revised and the format now more aligned with the reviewer’s suggestion.
  6. The US EPA (2009) and NESREA (2009) permissible levels of the metals are now included in Table one (1).
  7. Introduction
  8. The paragraph on Lines 53-69 has been revised to address the reviewer’s comments.
  9. Materials and Methods

     The figures were modified for greater clarity and visualization

  1. Lines 196-197: The number of embryos per treatment have now been added [90 embryos per treatment]
  2. All graphs have been checked and edited, as per the reviewer’s comment.
  3. Results
  4.    Added standardized letters of a and b to figure 5.
  5. Discussion
  6. As suggested, three new references have been added [i.e., 26, 46 and 47].
  7. Scientific names were checked and corrected throughout the manuscript for accurate spelling.